# Characteristics of Interfacial Shear Bonding Between Basalt Fiber and Mortar Matrix

**DOI:** 10.3390/ma13215037

**Published:** 2020-11-09

**Authors:** Li Hong, Tadan Li, Yadi Chen, Peng Gao, Lizhi Sun

**Affiliations:** 1Department of Structural Engineering, Hefei University of Technology, Hefei 230009, China; hongli2014@hfut.edu.cn (L.H.); tadanli@mail.hfut.edu.cn (T.L.); chenyadi@mail.hfut.edu.cn (Y.C.); 2Key Laboratory of Performance Evolution and Control for Engineering Structures of Ministry of Education, Tongji University, Shanghai 200092, China; 3Department of Civil & Environmental Engineering, University of California, Irvine, CA 92697, USA

**Keywords:** basalt fiber, interface, embedment length, pull-out test, shear bonding strength

## Abstract

Basalt fibers have been adopted as reinforcements to improve mechanical performance of concrete materials and structures due to their excellent corrosion resistance, affordable cost, and environmental-friendly nature. While the reinforcing efficiency is significantly dependent on fiber–matrix interfacial properties, there is a lack of studies focusing on the bonding behavior of basalt fibers in the mortar matrix. In this paper, a series of experiments were carried out to investigate the characteristics of single basalt fiber pulled out from the mortar matrix. Three embedment lengths and three types of mortar strength were considered. As references, the pull-out behavior of single polyvinyl alcohol (PVA) fiber and glass fiber in mortar matrix were also tested for comparison. Results from the pull-out test revealed that the average bonding strength is more effective than the equivalent shear bonding strength to illustrate the interfacial bond behavior of single basalt fiber in mortar matrix, which can be improved by either longer embedment length or the stronger mortar matrix. Finally, the tensile and compressive strengths of basalt/PVA/glass fiber-reinforced concrete (FRC) were measured to investigate the influence of interfacial shear bonding strengths. It was shown that, while PVA fiber developed the highest shear bonding strength with mortar, the basalt fiber exhibited the best reinforcing efficiency of FRC.

## 1. Introduction

Basalt fiber, as a natural material on earth, has recently received increasing attention for reinforcement in composites used in civil, aerospace, and electronic engineering. Basalt fiber is derived from volcanic rocks with a melting temperature between 1500 and 1700 °C [1,2]. In comparison to carbon and glass fibers, production of basalt fiber requires less energy without gas emission, reducing its environmental impact and cost significantly [3,4]. Moreover, when compared with the traditional fibers (e.g., steel, glass and carbon fibers), basalt fiber not only has an environmental-friendly manufacturing process [5], but also exhibits attractive properties, such as high tensile strength and elastic modulus [1], satisfactory adhesion strength [6], excellent acid and alkaline resistance [7], high thermal stability [2], and renewability [8]. In recent years, applications of basalt fiber in civil engineering have gained increasing interest, such as basalt fiber-reinforced polymer bars to substitute steel bars in reinforced concrete [9,10] and chopped fiber-filled cementitious materials [11,12]. Such a broad impact may be due to the fact that the natural basalt fiber requires low energy and few additives during production. Further, it is cost-effective and easy to disperse when mixed with mortar matrix, making the basalt fiber-reinforced concrete with decent workability and stability, excellent thermal resistance, and strong crack resistance and impact resistance.

Obviously, the efficiency of fiber reinforcement is strongly dependent on the fiber-bridging action which is controlled by the bonding property between the fiber and matrix [13,14,15]. Therefore, characteristics of fiber–matrix bonding property became a topic of interest. The authors of [13] concluded that pull-out test is an effective method to assess the interfacial bonding strength of the single fiber in cement matrix. The authors of [16] revealed that the bonding strength between steel fiber and mortar improves as the matrix strength increases. Conversely, the authors of [17] found that increasing both diameter and embedment length leads to higher bonding strengths of the interface between steel fiber and concrete matrix. This was confirmed by the authors of [18]. In addition to steel fibers, the interfacial bonding properties of polypropylene fiber, polyolefin fiber, coconut fiber, and hydrophilic PVA fiber, in cement matrix have been studied through pull-out tests in studies [19,20,21,22,23], respectively, and their results showed that (1) the interfacial bonding behavior is significantly affected by the matrix strength, fiber embedment length, fiber diameters, and pretreatment conditions; and (2) PVA fibers have both strong chemical and frictional bonding in the cementitious matrix [24].

For basalt fiber, while a few research efforts [15] were conducted on the larger scale on the bonding behavior of basalt fiber-reinforced polymer bars in cement matrix, the bonding strength between single basalt-fiber and cement matrix has not been studied in depth. The authors of [13] investigated the bonding strength of basalt fiber with cementitious material and revealed that basalt fiber has a strong chemical bonding with cementitious matrix before cracking process. Microstructure analysis was carried out [11,25], with the results indicating that basalt fiber has a good bonding with cement matrix. However, the effects including embedment length of single basalt fiber as well as the matrix strength were not considered in their work.

The current study therefore aims to characterize the bonding behavior of single basalt-fiber in the mortar matrix. First, a scanning electron microscope (SEM) and an energy dispersive X-ray (EDX) spectroscope are applied to study the fiber–mortar interface, in which, except basalt fiber, polyvinyl alcohol (PVA) fiber and glass fiber are also considered as references. Additionally, pull-out tests are carried on basalt/PVA/glass fiber with three different embedment lengths in three different strengths of mortar matrix. Consequently, the bonding behavior between mortar and basalt/PVA/glass fiber were analyzed and quantified using the initial load, maximum load, bond–slip curves, energy absorption, as well as shear bonding strength. Finally, the splitting tensile and compressive strengths of basalt/PVA/glass fiber-reinforced concrete (FRC) specimens are measured to investigate the influences of their interfacial shear bonding strength on the mechanical properties of FRC.

## 2. Research Significance

For the ongoing research [25], this work clarifies the pull-out behavior of basalt/PVA/glass fiber from mortar matrix, which is a depth research and supplementation for previous work. Moreover, it accesses the influence of fiber type, embedment length, mortar strength on pull-out behavior quantitatively using the initial load, maximum load, bond–slip curves, energy absorption, as well as direct shear bonding, which is limited in previous work. Furthermore, the study reported here was directed towards quantifying interfacial shear bond behavior and supplying mechanical data for the multiscale modeling and simulation of FRC materials and structures.

## 3. Material Design and Specimen Preparation

### 3.1. Materials and Mix Design

Short basalt fiber, PVA fiber, and glass fiber of 13 μm in diameter and 12 mm in length were prepared as shown in Figure 1, respectively. Additionally, three embedment lengths of basalt/PVA/glass fiber, i.e., 6, 9, and 12 mm, respectively, were employed for the pull-out test. The material properties of these fibers are shown in Figure 1, in which basalt fiber and glass fiber were supplied by Haining Composite Materials Company Ltd., Haining, China, while PVA fiber was supplied by Wanwei Group, Chaohu, China, of which the molecular formula is [C_2_H_4_O]_n_, and the Catalog number is W4. These fibers were provided by the manufacturing firms and directly used in the matrix without any preprocessing treatment. Ordinary Portland cement (42.5R), fresh water, cobblestones as coarse aggregates, and medium river sand (0.35–0.5 mm), of which the primary chemical composition is SiO_2_, were prepared for the fabrication of samples. The diameters of cobblestones were range from 5 mm to 20 mm, and the mass percentage is 23.8% (5–10 mm), 50.5% (10–16 mm) and 25.7% (16–20 mm), respectively.

Three different strengths of mortar matrix were investigated, referred to as M1, M2, and M3, all of which were produced using different mix proportions as listed in Table 1. Specifically, the water to cement ratio (W/C) of M1, M2 and M3 is 0.65, 0.49 and 0.40, respectively.

### 3.2. Specimens

To produce pull-out specimens, a special steel mold was designed as illustrated in Figure 2a, in which the distance between two closest steel sheets, *l*_e_, was set as 6, 9, and 12 mm, respectively, to realize and accept the different embedment lengths of fiber in cement matrix. To assure enough length inside the pull-out grip of the testing machine, the spacing between the two specimens was 50 mm. A hole with diameter of 2 mm was drilled from one side to the other for each steel sheet, as shown in Figure 2b, to allow the fiber to pass through.

The manufacture of a pull-out specimen can be achieved in the following manner:(1)The diameter of a single fiber is 13 μm, which is easily broken during the production. Therefore, a fiber bundle with a diameter of 1 mm was chosen and threaded through the round hole in steel sheets, which were then carefully placed in the module’s preset position (Figure 3a).(2)Both ends of the fiber were fixed to the mold using bolted joints.(3)The mortar was poured between the closest steel sheets (Figure 3b).(4)The specimens were removed from the mold approximately 24h after casting (Figure 3c) and cured in a conditioning chamber with a temperature of 20 ± 3 °C and relative humidity of 95 ± 5% for 28 days. All the work scopes (test factors, their differences, and number of specimens) for the fiber pull-out tests are summarized in Table 1.

For each strength of mortar matrix, six cubes with size of 70.7 mm were cast in order to assess their splitting tensile strength and compressive strength, according to the Standard for Test Methods for Mechanical Properties of Ordinary Concrete (GB/T 50081–2002 [26]). Furthermore, to investigate the effect of fiber–mortar bond property on the strength of FRC, for each type of fiber, six cubic specimens of FRC with each side of 100 mm were made. Seven normal concrete specimens were also produced as reference. For all these concrete specimens, the water/cement and cement/sand ratios were 0.40 and 0.84, which are the same as M3 listed in Table 1. The ratio between medium sand and coarse aggregates were 0.38. In FRC specimens, the volume fraction of fiber was 1.0%. Finally, all specimens were cured under conditions at 25 ± 2 °C and 100% R.H. for 28 days. They were then tested by a hydraulic pressure system.

For the microstructure between fiber and mortar matrix, one of FRC specimens were cut into slices with 1 mm thick or less using a diamond saw to expose fiber–mortar interfaces, then the surfaces of slices were coated with gold for SEM and EDX analysis.

#### Experimental Methods

The pull-out test was conducted on a computer-controlled universal testing machine with a special designed metallic frame for loading, as depicted in Figure 4. First, the free end of the fiber in the specimen was threaded through the round hole of the steel sheets and then secured on the end using the epoxy to keep the fiber from slipping back through. Test was conducted based on protocol established in [22,24], and controlled by the position of the actuator, which had a displacement rate of 1 mm/min. The displacement and force during the test can be recorded by a data acquisition system, as shown in Figure 4. The compressive strength and the splitting tensile strength of mortar and concrete specimens were determined by a 3000-kN-capacity pressure testing machine.

## 4. Results and Discussion

### 4.1. Mechanical Properties of Mortar Matrix

For the mortar with a W/C ratio of 0.65, 0.49, and 0.4, the average compressive strength of three mortar specimens was 27.7, 40.2, and 49.9 MPa, respectively, and the average splitting tensile strength was 3.28, 4.26, and 6.06 MPa, respectively. This is in line with the conclusion from [25] that the strength of mortar can be improved by the lower of W/C ratio.

### 4.2. Microstructures of Interface Between Basalt/PVA/Glass Fiber and Mortar Matrix

Figure 5 presents typical SEM micrographs of fiber–mortar interface for basalt, PVA, and glass fiber, respectively. It can be observed in Figure 5a that the interface between basalt fiber and mortar is full of the hydration product, i.e., calcium silicate hydrate (C-S-H), without voids and cracks. Figure 5b shows a sound interface between PVA fiber and mortar matrix. However, the interface between glass fiber and mortar presented in Figure 5c has microcracks.

EDX analysis results in Figure 6 show that the Ca/Si ratios of basalt, PVA and glass fiber–mortar interface are 1.47, 1.30 and 2.44, respectively, indicating that basalt fiber–mortar interface has a lower amount of hydration products (e.g., C-S-H) than PVA fiber–mortar interface, yet having a higher amount of hydration products than glass fiber–mortar interface. This agrees with our previous work in [25], since basalt fibers have stronger chemical bonding strength with mortar matrix due to their similar chemical composition such as SiO_2_ [13], whereas PVA fibers are hydrophilic which results in hydrophilic interactions between PVA fiber and mortar matrix [24]. However, glass fiber without any surface preprocess used in this work is hydrophobic [27].

### 4.3. Pull-Out Behavior of Basalt/PVA/Glass Fiber in Mortar Matrix

#### 4.3.1. Failure Modes of Fiber Pull-Out

Figure 7 presents two typical failure modes of fiber pull-out: the fiber with complete pull-out from mortar (Figure 7a) and the ruptured fiber (Figure 7b). Clearly, the basalt, PVA, and glass fiber in Figure 7a are fully pulled out from the mortar, accompanied by a photo of the broken mortar wedge (particularly for the PVA fiber) occurred during the test. Figure 7b depicts the ruptured basalt, PVA, and glass fiber, one part of which embedded in mortar, and another is in steel sheets.

#### 4.3.2. Maximum Load in Pull-Out Test

For the specimens in which fibers are fully pulled out from the mortar matrix, the maximum pull-out loads of basalt, PVA and glass fibers are exhibited in Figure 8a–c, respectively, in which the number of scatter data represents the number of failed specimens.

First, it can be found in Figure 8 that the number of specimens in which fiber was successfully pulled out decreases with an increase in the embedment length of fiber. For example, all specimens are failed by fiber fully pulled out when the embedment length is equal to 6 and 9 mm. However, for the embedment length of 12 mm, only one or two specimens failed by fiber fully pulled out. Similar results were also obtained in studies of [23,28]. This may be due to the higher pull-out load than fiber tensile load and the area defining the fiber cross-section dimension along the fiber length being smaller during the pull-out process.

Additionally, Figure 8 shows that the maximum load is significantly enhanced by the longer embedment length of the fiber. Specifically, for specimens with a W/C ratio of mortar equal to 0.49, the maximum load of basalt, PVA, and glass fiber is increased by 34.6%, 94.9%, and 33.8%, respectively, when the embedment length of fiber changed from 6 mm to 9 mm. Moreover, they are increased by 63.3% for basalt fiber, 170.7% for PVA fiber, and 65.8% for glass fiber when the embedment length of fiber changed from 6 to 12 mm. This finding is consistent with that from [28], which could be attributed to the increased interfacial area between fiber and the mortar.

Furthermore, all maximum loads present a significant decrease with an increase in the W/C value of mortar matrix. For example, when the value of W/C increases from 0.40 to 0.49 for specimen with a fiber embedment length equal to 9 mm, the maximum load of basalt, PVA, and glass fiber decreases by 4.9%, 13.5%, and 6.7%, respectively. When the value of W/C increases from 0.40 to 0.65 for the 9 mm fiber, the maximum load of basalt, PVA, and glass fiber decreases by 25.8%, 38.0%, and 27.0%, respectively. This is in agreement with other results reported previously in [17,23,29]. This pronounced decrease may be caused by the stronger bond between fiber and stronger mortar when the W/C value increased from 0.40 to 0.65.

It is also noted that the maximum load is obviously affected by the type of fiber. As illustrated in Figure 8, for the specimen with same fiber embedment length and mortar strength, basalt fiber exhibits a lower maximum load than PVA fiber, but a higher load than glass fiber. Such behavior can be explained by the microstructure of interface between fiber and mortar: (1) a lot of microcracks in the glass fiber–mortar interface significantly reduced the adhesion between glass fiber and mortar (as shown in Figure 5c); and (2) PVA fiber–mortar interface exhibits the highest amount of hydration products, followed by basalt fiber, and then the glass fiber (as shown in Figure 5a–c). More hydration products can effectively improve the interfacial bond.

#### 4.3.3. Representative Load–Slip Curves of Fiber

The typical load–slip curves obtained from the pull-out tests of basalt, PVA, and glass fibers embedded in mortar matrix are shown in Figure 9, Figure 10 and Figure 11, respectively. To focus solely on pull-out behavior, the curves are prepared by subtracting the initial slope obtained from the elastic deformations. Therefore, the curves starting point represents where the fibers first started debonding.

Obviously, all typical load–slip curves of fiber in Figure 9, Figure 10 and Figure 11 are characterized by a rapid increase of load until the maximum load is reached, followed by a drop as the fiber is debonded. After this, only frictional resistance remains until the fiber fully pulls out of the mortar matrix. The representative values of the pullout load can be identified as the initial load that represents where the fiber debonding took place [22], and the maximum load means the fiber was completely debonded from the mortar matrix.

It can first be observed in Figure 9, Figure 10 and Figure 11 that the maximum loads show a growing trend with an increase in the fiber embedded length and the strength of mortar matrix, which has been clarified above.

Moreover, the initial loads for basalt, PVA, and glass fiber from the mortar matrix presents in Figure 9, Figure 10 and Figure 11 are significantly different. Notably, the initial load experiences an incremental change when the fiber embedment length became longer, or the mortar matrix became stronger. The main reason for the enhanced initial load is due to the larger area caused by the longer fiber and improved matrix packing density around the fiber, leading to a higher bond between the fiber and mortar matrix.

Additionally, the initial stiffness obtained from the linear portion’s slope of the load–slip curve exhibits a similar trend as the initial load, which is also improved by the longer fiber embedment length and the higher strength of mortar matrix. However, with the same mortar matrix and fiber embedded length, PVA fiber always appears with a higher maximum load than either basalt fiber or glass fiber. This difference is most likely due to the highest amount of hydration products in the interface between PVA fiber and mortar [24], which can effectively improve their adhesion. Another significant difference observed in Figure 9, Figure 10 and Figure 11 is that, after the maximum load is reached, glass fibers experience a sharp drop. This is not the case with the basalt or PVA fiber showing a higher frictional bonding resistance than glass fiber. Similar phenomena were also obtained from [20,24].

For an in-depth analysis, the energy absorption during the pull-out is calculated as the area under the load-slope curve. The influences of fiber embedment length and mortar matrix strength on the energy absorption are illustrated in Figure 12. Both reveal that basalt fiber absorbed less energy than PVA fiber, but more than glass fiber during pull-out testing. This is confirmed with the above findings that basalt fiber showed lower maximum load and frictional bonding resistance than PVA fiber, but higher than glass fiber. The main reason can also be attributed to the sounder microstructure and more hydration products in the PVA fiber–mortar interface [24].

Figure 12 also shows that the longer embedment length of fiber significantly improves the absorbed energy, whereas the W/C of mortar matrix has less effect. This difference may be due to the fact that: (1) longer embedment length of fiber increases the area of pull-out load–slip curve, which need more energy for the fiber fully pulled out; (2) mortar matrix providing the primary chemical bond for the fiber–mortar interface, while it has less effect on the frictional bond of the interface. Therefore, it can be concluded that fiber embedment length has a greater effect on the absorbed energy than the mortar strength in the pull-out test.

#### 4.3.4. Interfacial Bonding Strength Between Fiber and Mortar Matrix

Based on the works of [13,22,24], the shear bonding strength of the interface between fiber and matrix can be divided as the average shear bonding strength and the equivalent shear bonding strength.

The average shear bonding strength *τ*_sa_, which based on the maximum load *P* and the embedment length *l*_e_, is given by:(1)$$\tau_{\mathrm{sa}}=\frac{P}{\pi \cdot d_{f}\cdot l_{e}}$$
where *d*_f_ is the diameter of the fiber. However, the equivalent shear bonding strength, *τ*_se_, which based on the absorbed energy *E,* can be calculated by:(2)$$\tau_{\mathrm{se}}=\frac{2E}{\pi \cdot d_{f}\cdot {l_{e}}^{2}}$$

Figure 13 illustrates the influences of mortar strength and fiber embedment length on the average shear bonding strength of the interface. It is shown that, at a given embedment length, all fibers possess a higher *τ*_sa_ with a decreasing W/C of the mortar matrix. For the basalt fiber with embedment length equals to 6 mm, when the W/C of the mortar matrix decreased from 0.65 to 0.49 or from 0.65 to 0.40, there was an increase of around 28% and 37% in τ_sa_, respectively. The corresponding increase for PVA fiber was 39% at 0.49 WC and 61% at 0.40 W/C. For glass fiber, it was 28% at 0.49 and 35% at 0.40 with both W/C values of mortar matrix decreased from their starting point of 0.65. This is consistent with the results of [30], and can be explained since higher strength of mortar matrix with a lower W/C value improves the interfacial bonding properties.

Figure 13 compares the given embedment lengths of fiber and W/C values of mortar matrix with all three of the featured fibers. Therefore, the basalt fiber presents a lower τ_sa_ than PVA fiber, but slightly higher τ_sa_ compared to that of glass fiber. However, Figure 13 also reveals that the influence of embedment length of fiber on τ_sa_ is significantly different from that of the W/C value. With the increase of embedment length, basalt fiber and glass fiber exhibit a typical decline in τ_sa_, whereas PVA fiber shows a gradual increase of τ_sa._ That is to say, the average shear bonding strength between fiber and mortar matrix does not always increase with a longer fiber embedment length, even if the maximum load increased. This is due to the fact that a longer embedment length (*l*_e_) not only causes a larger maximum load, which is beneficial to τ_sa_, but an adverse effect also appears on τ_sa_ as illustrated in Equation (1). For PVA fiber, the adverse effect caused by the longer embedment length of fiber on τ_sa_ is lower than its benefit. Therefore, the value of τ_sa_ for PVA fiber shows an increase. However, for basalt fiber and glass fiber, the longer embedment length of fiber has a more significant adverse effect, which results in a decrease of τ_sa._ Previous research on pull-out glass fiber embedment length effects and maximum loads showed similar results [31].

In addition, the relation between fiber embedment length, mortar matrix strength and equivalent shear bonding strength *τ*_se_ for each type of fiber is depicted in Figure 14. Clearly, basalt fiber presents a markedly lower *τ*_se_ value than PVA fiber, but a higher of that than glass fiber. This higher *τ*_se_ value also can be explained by the highest amount of hydration products between PVA fibers and mortar matrix, followed by basalt fibers, and then glass fibers [24]. The effects of fiber embedment length and the W/C ratio of mortar matrix on the value of *τ*_se_ are also significant even though no rule can be observed. However, test values are different when using steel fibers, which exhibit a typical growth with increasing concrete compressive strength in the study of [30].

Above all, it can be concluded that the average shear bonding strength is more effective than the equivalent shear bonding strength when considering the interface between flexible fiber and mortar matrix, since the average shear bonding strength changes regularly based on different fiber embedment lengths and the W/C ratios of mortar matrix. It is thus important to note the different types of fiber and their different pull-out test response patterns.

### 4.4. Effect of Interfacial Bonding Property on Mechanical Properties of FRC

Above all, basalt fiber, PVA fiber and glass fiber showed different interfacial bonding properties, which can undoubtedly lead to different levels of strength for fiber-reinforced concrete. Figure 15 shows the splitting tensile strengths and compressive strengths of normal concrete (NC), basalt fiber-reinforced concrete (BFRC), PVA fiber-reinforced concrete (PFRC), and glass fiber-reinforced concrete (GFRC), respectively. As can be seen in Figure 15a, PFRC exhibits 26.4% higher splitting tensile strength than that of NC, followed by BFRC, which shows 14.6% higher, whereas GFRC almost has no change. Both the differences in the interfacial bonding strength between fiber and mortar matrix and the fibers’ mechanical properties were considered reasons for this effect. However, the influence of fiber’s mechanical properties could be negligible, since it was observed that most of the fibers were fully pulled out from the mortar matrix. Therefore, the interfacial bonding strength is thought to be the main reason. As previously mentioned, PVA fiber provides the strongest interfacial bonding strength, followed by basalt fiber, and then glass fiber, which results in a similar decrease in the splitting tensile strength of PFRC, BFRC and GFRC. Here, the micromechanical model developed by [32] can be adopted to link the interfacial properties from previous section with the compressive strength. However, some parameters, such as crack length, fracture toughness of cementitious materials without fibers, and coefficient of friction against shear sliding of crack faces, cannot be determined by our existing experiments. Future research will be needed to quantify and verify such link between interfacial properties and effective mechanical responses of FRC.

However, it can be obtained from Figure 15b that, when compared with NC, the compressive strength of BFRC almost unchanged, whereas the compressive strength of PFRC and GFRC decreased by 25.7% and 16.9%, respectively. Such large decrease may be due to the fact that vibrations of PFRC were not enough and the number of pores was increased, causing the density worse after adding short PVA fibers into concrete [33]. Undoubtedly, the large number of initial microcracks in the glass fiber–mortar interface is the main reason for the degradation of GFRC’s compressive strength.

From the results in Figure 15a,b, it can be concluded that basalt fiber is more effective than either PVA fiber or glass fiber to improve the mechanical properties of concrete, since it can improve the tensile strength of concrete as well as keep their compressive strength same.

## 5. Conclusions

This paper aims to systematically investigate and clarify the pull-out behavior of single basalt fiber in mortar matrix, in which the influences of fiber embedment length and matrix strength are taken into consideration. The pull-out behavior is characterized and quantified with mechanical parameters, including maximum load, initial load, initial stiffness, energy absorption, and shear bonding strength. Moreover, the same parameters of single glass fiber and PVA fiber, which serve as references, are also analyzed. According to the experimental results and discussion, the following conclusions can be obtained:The maximum load, initial load, initial stiffness, energy absorption, and average bonding strength for the basalt fiber pulled out from mortar matrix, are significantly improved with fiber’s longer embedment length and the lower W/C of mortar matrix.The average bonding strength is more effective than the equivalent shear bonding strength to demonstrate the interfacial bond behavior of single basalt fiber in mortar matrix.The interfacial bonding strength between single basalt fiber and mortar matrix is higher than glass fiber, but lower than PVA fiber.Basalt fiber presents a greater improvement than PVA fiber or glass fiber in FRC materials.The obtained results provide a foundational basis of mechanical data for the multiscale modeling and simulation of FRC materials and structures.

## Figures and Tables

**Figure 1 materials-13-05037-f001:**
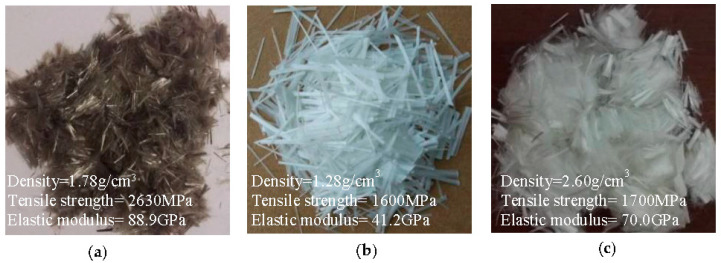
Three types of fiber: (**a**) basalt fiber, (**b**) polyvinyl alcohol (PVA) fiber, and (**c**) glass fiber.

**Figure 2 materials-13-05037-f002:**
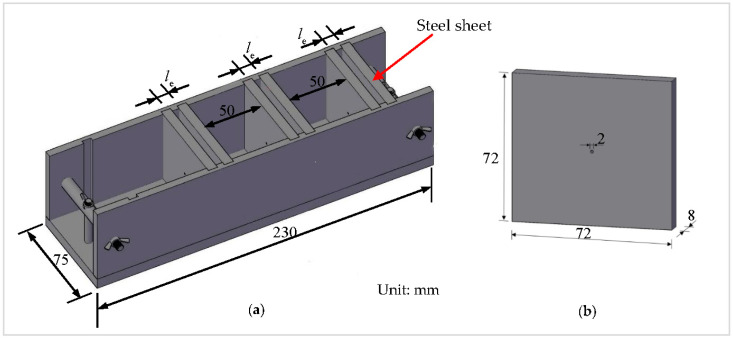
The steel mold of the fiber pull-out specimen: (**a**) steel mold, and (**b**) steel sheet.

**Figure 3 materials-13-05037-f003:**
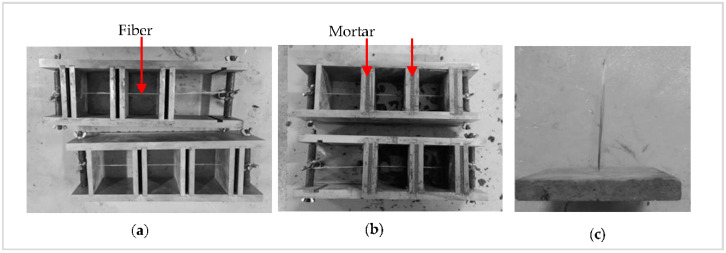
The procedures of fiber pull-out specimen: (**a**) threading the fiber, (**b**) casting the mortar matrix, and (**c**) the pull-out specimen.

**Figure 4 materials-13-05037-f004:**
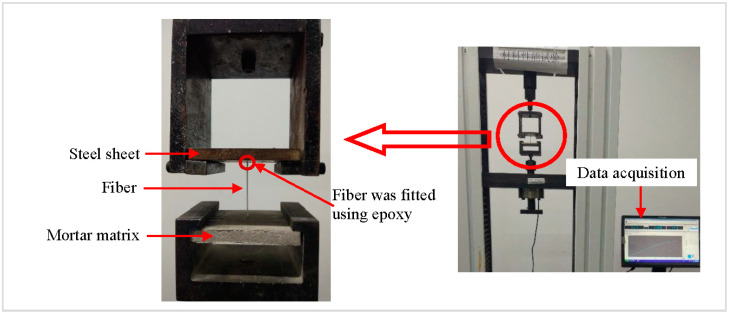
The pull-out test setup.

**Figure 5 materials-13-05037-f005:**
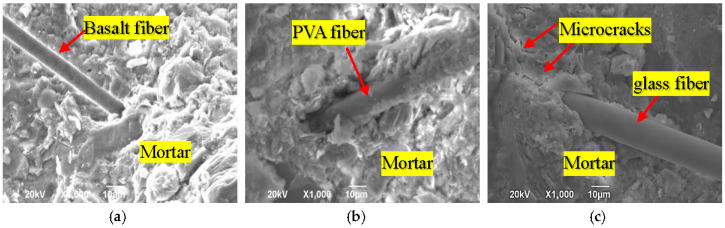
Interface between mortar and: (**a**) basalt fiber, (**b**) PVA fiber, and (**c**) glass fiber.

**Figure 6 materials-13-05037-f006:**
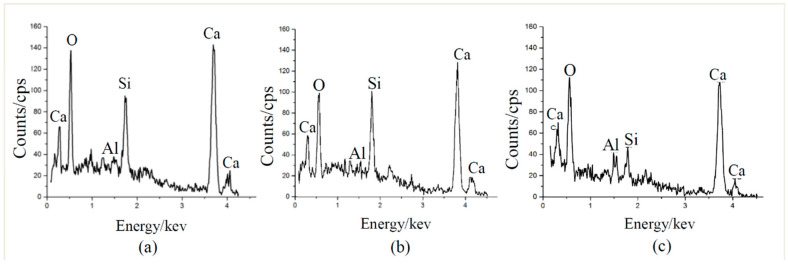
EDX results of (**a**) basalt fiber-mortar interface, (**b**) PVA fiber-mortar interface, and (**c**) glass fiber-mortar interface.

**Figure 7 materials-13-05037-f007:**
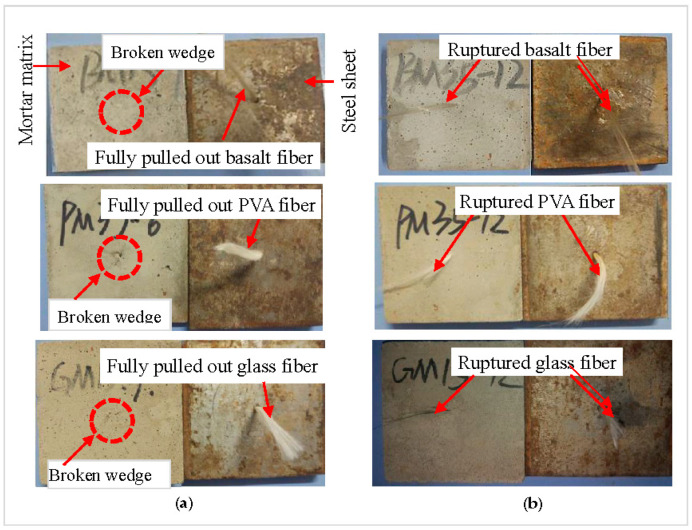
Typical failure modes of pull-out specimens: (**a**) fiber fully pulled out from mortar, and (**b**) fiber rupture.

**Figure 8 materials-13-05037-f008:**
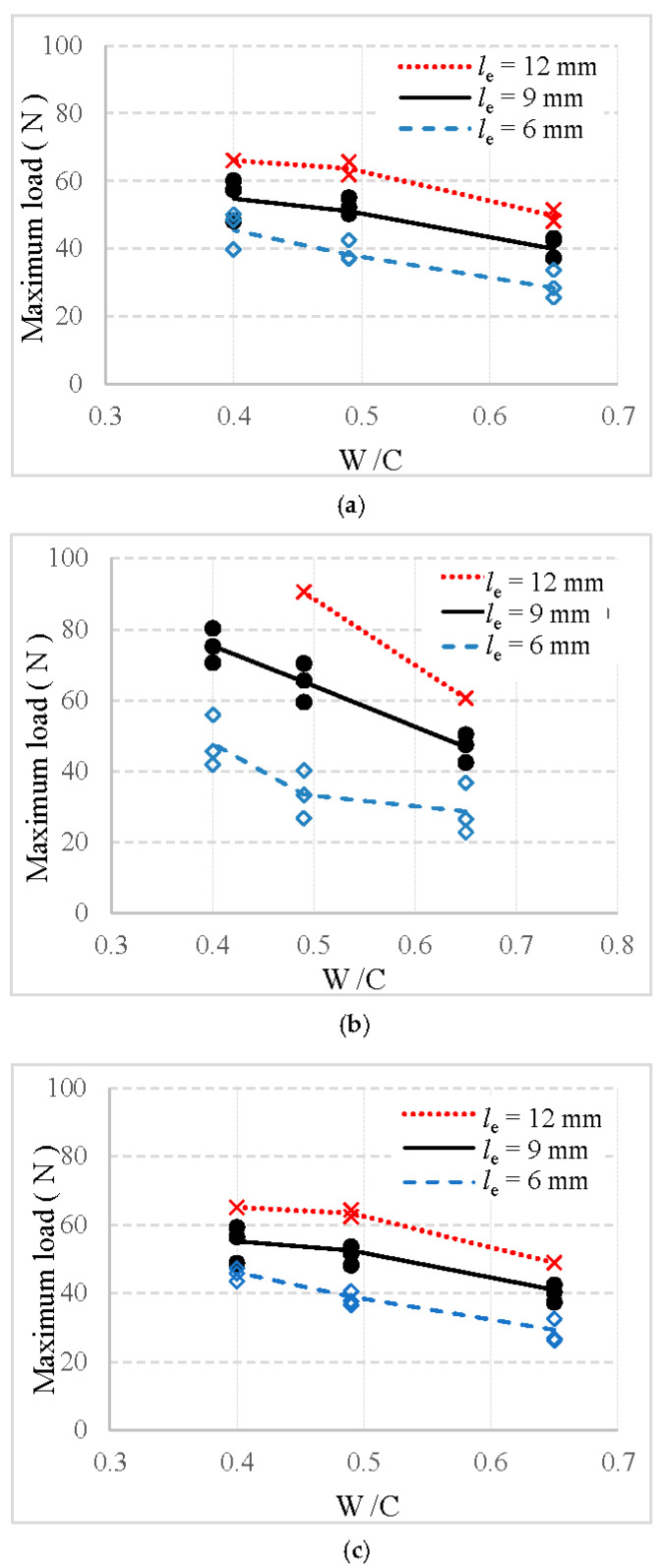
Maximum loads of fiber pulled out from mortar: (**a**) basalt fiber, (**b**) PVA fiber, and (**c**) glass fiber.

**Figure 9 materials-13-05037-f009:**
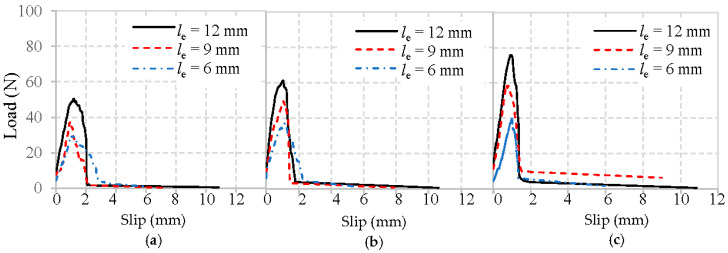
Load–slip curves of basalt fibers pulled out from the mortar matrix: (**a**) W/C = 0.65, (**b**) W/C = 0.49, and (**c**) W/C = 0.40.

**Figure 10 materials-13-05037-f010:**
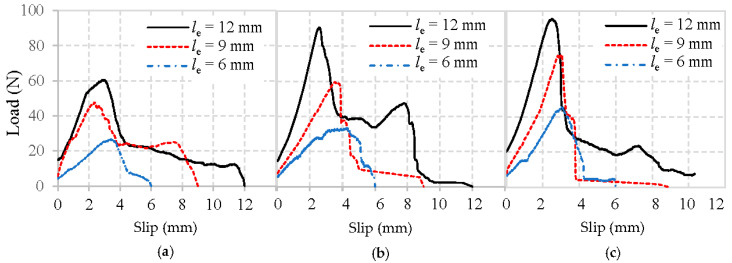
Load–slip curves of PVA fibers pulled out from the mortar matrix: (**a**) W/C = 0.65, (**b**) W/C = 0.49, and (**c**) W/C = 0.40.

**Figure 11 materials-13-05037-f011:**
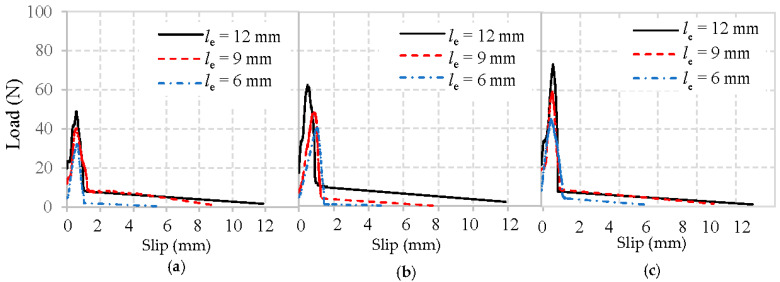
Load–slip curves of glass fibers pulled out from mortar matrix: (**a**) W/C = 0.65, (**b**) W/C = 0.49, and (**c**) W/C = 0.40.

**Figure 12 materials-13-05037-f012:**
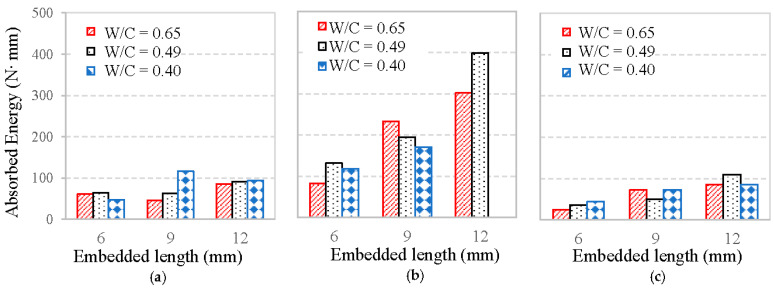
Energy absorption: (**a**) basalt fiber, (**b**) PVA fiber, and (**c**) glass fiber.

**Figure 13 materials-13-05037-f013:**
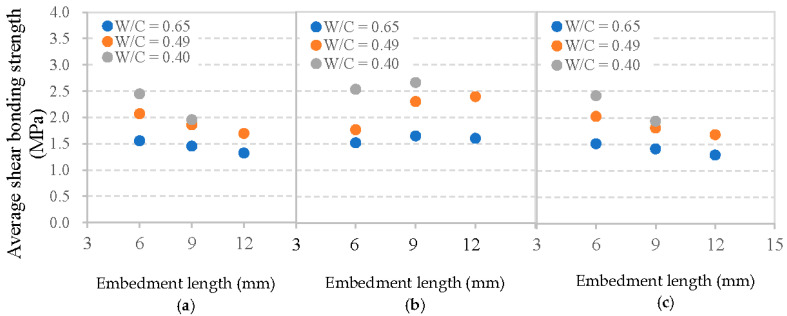
Average shear bonding strength between cement matrix and: (**a**) basalt fiber, (**b**) PVA fiber, and (**c**) glass fiber.

**Figure 14 materials-13-05037-f014:**
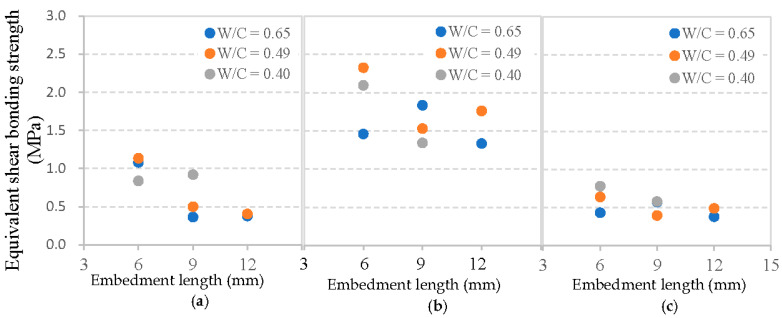
Equivalent shear bonding strength between cement matrix and: (**a**) basalt fiber, (**b**) PVA fiber, and (**c**) glass fiber.

**Figure 15 materials-13-05037-f015:**
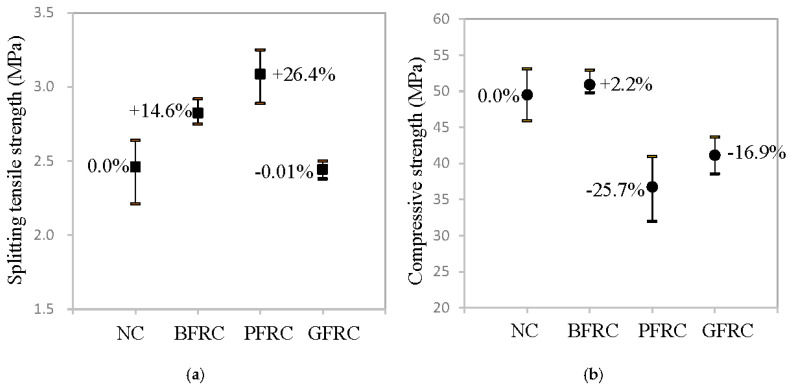
Test results of concrete specimens: (**a**) splitting tensile strength; and (**b**) compressive strength.

**Table 1 materials-13-05037-t001:** The work scope of the pull-out test specimens.

Test Factors	Cases	Specimens’ Number
Fiber type	(a) Basalt fiber	3
	(b) PVA fiber	3
	(c) Glass fiber	3
Fiber embedment length (mm)	(a) 6	3
	(b) 9	3
	(c) 12	3
Mix proportions of mortar matrix (water ***: cement: sand) (kg/m^3^)	(a) 0.65: 1: 2.4 (M1)	3
	(b) 0.49: 1: 1.5 (M2)	3
	(c) 0.40: 1: 1.2 (M3)	3

*: not included the water absorbed by the fibers.

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
