# Peer review of "Characteristics of Interfacial Shear Bonding Between Basalt Fiber and Mortar Matrix"

_materials, 2020, doi:10.3390/ma13215037_

Round 1

Reviewer 1 Report

Characteristics of interfacial shear bonding between basalt fiber and mortar matrix

Li Hong1,2,3, Tadan Li1, Yadi Chen1, Peng Gao1*, Lizhi Sun3*

1Department of Structural Engineering, Hefei University of Technology, Hefei 230009, P.R. China

2Key Laboratory of Performance Evolution and Control for Engineering Structures of Ministry of China, Tongji University, Shanghai 200092, P.R. China

3Department of Civil & Environmental Engineering, University of California, Irvine, CA 92697, USA

A series of experiments were done to test the characteristics of single basalt fiber pulled out from the mortar matrix. For comparison PLA and fiberglass fiber in mortar matrix were also tested for comparison. Authors also completed tensile and compressive strengths measurements of basalt/PVA/glass fiber-reinforced concrete (FRC). Their results show the basalt fiber exhibited the best reinforcing efficiency of FRC.

The publication has few typing errors, see line 37: increasing interest instead of increasingly interest, line 89: 13μm instead of 13 μm, line 193: (a) basalt fiber,.(b) PVA fiber instead of (a) basalt fiber, (b) PVA fiber.

PVA is poorly characterized – Mn, MWD, -OH %, crystallinity. Catalog number and the name of the manufacturer would also be sufficient.

Splitting tensile strength usually done on a cylindrical specimen and not on a cubic sample.

line 175: Glass fiber is not hydrophobic – the surface of a glass is covered by Si-OH chemical groups. (I have not checked the literature cited.)

lines 362-370: There is a discrepancy between lines 362-363 and line 370. 2,2% is not a real increase, as it stated later: “it can…….keep their compressive strength same.”

Reviewer 2 Report

The paper is quantifying mechanical properties of the interface between basalt fiber and mortar. The study can be useful in terms of supplying mechanical data for the multiscale modelling and simulation of FRC materials and structures.

The paper is well written with vey good English standard - although further proof reading is highly recommended.

In Table 1, different ratio of (water: cement: sand) were used for the mortar matrix. Does this ratio considered the effect of water absorption properties of different fibres? 

How you made sure that the epoxy used (to keep the fiber from slipping back through) did not influence the results?

The source and (chemical composition) of the sand used should presented and discussed in terms of its effect on the bonding behaviour between fiber and mortar.

Reviewer 3 Report

The paper presents a study on the interfacial shear bonding between
basalt fiber and mortar matrix.
The manuscript shows some (limited) experimental results from an ongoing research project of the same authors. From the same project, other results are presented in https://doi.org/10.1016/j.conbuildmat.2019.117235 (2020).

While the research topic is certainly of interest and should be considered for publication, this document seems a short derivation that is much weaker than the other journal paper. I mean that I would expect the same rigorous analysis and discussion of results in both the documents. What I see in this manuscript under review is now a paper content that should be further discussed and improved in its presentation.

  • given that "six cubes" are used in the study, there's no need to write 70.7 mm ×70.7 mm ×70.7 mm but it is clear that all the edges have the same size
  • please improve the graphical quality of fig. 6, it is hard to read
  • the test setup and instruments seem the same from the other paper, https://doi.org/10.1016/j.conbuildmat.2019.117235. If so, there's no more reason to describe them again
  • figure 5 of ths manuscript has strong similarity with figure 3 in https://doi.org/10.1016/j.conbuildmat.2019.117235. I do not say that the content is exactly the same (no plagiarism), but I do not see new content in this figure
  • comparison of the current test results and possible literature studies are missing. The paper would take benefit of this
  • in conclusion, I cannot justify (in this current form) a new journal publication for this study. The impact is limited and strictly related to the other publlication from the same authors. This manuscript could be re-considered for publication once it is properly modified in content, significance, presentation

Round 2

Reviewer 3 Report

Some improvements are added in the revised paper.